# Fault Diagnosis of Wind Turbine Gearbox Based on Improved Multivariate Variational Mode Decomposition and Ensemble Refined Composite Multivariate Multiscale Dispersion Entropy

**DOI:** 10.3390/e27020192

**Published:** 2025-02-13

**Authors:** Xin Xia, Xiaolu Wang, Weilin Chen

**Affiliations:** 1School of Mechanical and Electrical Engineering, Suqian University, Suqian 223800, China; wangxiaolu@squ.edu.cn; 2BLUE.x.y Intelligent Technology Co., Ltd., Suqian 223800, China; weilinchenbluexy@gmail.com

**Keywords:** multivariate variational mode decomposition, refined composite multivariate multiscale dispersion entropy, fault diagnosis, planetary gearbox

## Abstract

Wind turbine planetary gearboxes have complex structures and operating environments, which makes it difficult to extract fault features effectively. In addition, it is difficult to achieve efficient fault diagnosis. To improve the efficiency of feature extraction and fault diagnosis, a fault diagnosis method based on improved multivariate variational mode decomposition (IMVMD) and ensemble refined composite multivariate multiscale dispersion entropy (ERCmvMDE) with multi-channel vibration data is proposed. Firstly, the IMVMD is proposed to obtain the optimal parameters of the MVMD, which would make the MVMD more effective. Secondly, the ERCmvMDE is proposed to extract rich and effective feature information. Finally, the fault diagnosis of the planetary gearbox is achieved using the least squares support vector machine (LSSVM) with features consisting of ERCmvMDE. Simulations and experimental studies indicate that the proposed method performs feature extraction well and obtains higher fault diagnosis accuracy.

## 1. Introduction

Planetary gearboxes are common transmission components with a large carrying capacity, high transmission ratio, and balance torque. They have been widely used in wind turbines. As the operating conditions of a planetary gearbox are often severe, with harsh environments, there will be frequent damage to its internal components [1]. Failures of the planetary gearbox will influence the transmission efficiency and may even lead to serious damage to the entirety of the equipment [2]. Therefore, the effective fault diagnosis of planetary gearboxes in wind turbines is very important.

Existing fault diagnosis methods mainly focus on the feature extraction of the vibration signals of planetary gearboxes, and the effectiveness of feature extraction directly influences fault diagnosis [3,4]. Owing to the complex structure of the planetary gearbox, the vibration signals of various components are always coupled and modulated with each other, and other factors also influence the vibration signal, such as background noise [5,6]. Fault feature extraction with the original vibration signals usually does not result in good performance. It is necessary to preprocess the original vibration signals for fault feature extraction.

The vibration signals of planetary gearboxes are usually nonlinear and nonstationary, and empirical mode decomposition (EMD) can decompose the different modal and noise components automatically [7]. Variational mode decomposition (VMD) has been proposed to overcome the problems of endpoint effects and mode mixing of EMD [8], and VMD has been widely used in the signal processing of planetary gearboxes [9,10]. Some studies indicate that the intensity and spectral structure of vibration signals are not the same for different fault locations, which may cause misjudgment when using a single-direction vibration signal. In recent years, an increasing number of studies have used multi-channel vibration data for fault diagnosis, and the multivariate signal analysis method ensures that each channel has the same decomposition scale; this has been widely used in planetary gearbox fault diagnosis [11]. Rilling et al. extended EMD to bivariate empirical mode decomposition (BEMD) [12]. Rehman and Mandic proposed multivariate empirical mode decomposition (MEMD) to process multi-channel data [13]. Moreover, multivariate variational mode decomposition (MVMD) was proposed by Rehman and Aftab to overcome the problems of endpoint effects and mode mixing [14]. However, the preset parameters of MVMD have a significant influence on its processing results [15], and parameter optimization is still an important issue.

Many nonlinear dynamic methods, such as sample entropy (SE), permutation entropy (PE), fuzzy entropy (FE), dispersion entropy (DE), and their multiscale entropy forms, have been widely used in the field of planetary gearbox fault diagnosis because of their ability to extract nonlinear fault characteristic information hidden in vibration signals [16,17,18,19,20]. Owing to the richer fault feature information contained in multivariate multiscale entropy, various methods have also been proposed and applied, such as mvMSE, mvMFE, and mvMDE [21,22,23].

Multiscale entropy may lead to inaccurate calculations owing to sequence coarse-graining when the sample is too short and it cannot adequately extract the fault features [24]. Refined composite methods, such as RCMSE, RCMFE, and RCMDE, have been proposed to improve the computational stability of multiscale entropy [25,26,27]. Refined composite multiscale entropy has been applied in the field of planetary gearbox fault diagnosis and has yielded many research results [28,29]. Refined composite multiscale entropy generally obtains a coarsened sequence using the mean value. However, Azami H et al. obtained the coarsened sequence by variance and standard deviation, and the results indicated that RCMFE based on different coarse-graining methods can obtain different features at different times [27]. Therefore, research on refined composite methods for multiscale entropy is important for the effective extraction of fault features.

Based on the above analysis, a fault diagnosis method based on improved multivariate variational mode decomposition (IMVMD) and ensemble refined composite multivariate multiscale dispersion entropy (ERCmvMDE) with multi-channel vibration data is proposed for wind turbine planetary gearboxes. The main contributions of this study are as follows. A comprehensive evaluation index based on orthogonality and dispersion entropy is proposed to obtain the optimal parameters of the MVMD. An ensemble refined composite multivariate multiscale dispersion entropy is proposed to form the composite feature. The fault diagnosis of the wind turbine planetary gearbox is achieved with a least squares support vector machine (LSSVM) using the proposed feature extraction method.

The remainder of this paper is organized as follows. The related theory, proposed signal processing method, and proposed feature extraction method are introduced in Section 2. In Section 3, the proposed fault diagnosis method is described. In Section 4, simulation analyses are presented to verify the advantages of the proposed IMVMD and ERCmvMDE. In Section 5, fault diagnosis using the proposed method is analyzed and compared. Finally, the conclusions are drawn in Section 6.

## 2. Methods

### 2.1. Improved Multivariate Variational Mode Decomposition

#### 2.1.1. The Basic Theory of MVMD

MVMD is a multi-channel signal processing method that evolved from VMD [14]. MVMD can perform the coordinated decomposition of multi-channel signals.

Assume that a multi-channel signal $S={\left\{s_{p}(t)\right\}}^{p=1,2,\ldots ,P}$, which contains *P* channels, can be decomposed into a series of intrinsic mode functions (IMFs) by MVMD as(1)$$S(t)=\sum_{k=1}^{K}u_{k}(t)$$
where *K* is the number of IMFs, $u_{k}(t)=\left[u_{1}(t),u_{2}(t),\ldots ,u_{P}(t)\right]$.

MVMD requires that the sum of the bandwidth of the IMFs be the minimum. The original signal can be accurately reconstructed by IMFs simultaneously. The constrained problem is expressed as follows:(2)$$\left\{\begin{array}{l} \underset{\left\{u_{p,k}\right\},\left\{\omega_{k}\right\}}{\min}\left\{\sum_{k}\sum_{p}{\left\|\partial_{t}\left[u_{+}^{k,p}(t)e^{-j\omega kt}\right]\right\|}_{2}^{2}\right\}\\ s.t.\;\sum_{k}u_{p,k}(t)=s_{p}(t),\;p=1,2,\ldots ,P \end{array}\right.$$
where $\partial_{t}[\cdot ]$ represents the calculation of the partial derivatives of time, and $u_{+}^{k,p}(t)$ is the modulating signal of the *k*-th IMF of the *p*th signal channel. $\omega_{k}$ is the center frequency of the IMF. The bandwidth of $u_{k}(t)$ can be estimated by the *L*_2_ norm of the gradient function of $u_{+}^{k}(t)$.

To solve the problem in Equation (2), its corresponding augmented Lagrangian function can be expressed as(3)$$\begin{array}{l} L(\{u_{k,p}\},\{\omega_{k}\},\lambda_{p})=\alpha {\sum_{k}\sum_{p}\left\|\partial_{t}[u_{+}^{k,p}(t)e^{-j\omega kt}]\right\|}_{2}^{2}+\sum_{p}{\left\|s_{p}(t)-\sum_{k}u_{k,p}(t)\right\|}_{2}^{2}+\\ \sum_{p}\langle \lambda_{p}(t),\;s_{p}(t)-\sum_{k}u_{k,p}(t)\rangle \end{array}$$
where *α* is the penalty factor, *λ_p_*(*t*) represents the Lagrange multiplier, and <·,·> represents the inner product.

The alternate direction method of multipliers (ADMM) is used to obtain the optimal solution of MVMD for Equation (3), where the modal update is as follows:(4)$${\hat{u}}_{k,p}^{n+1}(\omega )=\frac{{\hat{s}}_{p}(\omega )-\sum_{i\neq k}{\hat{u}}_{i,p}(\omega )+\frac{{\hat{\lambda }}_{p}(\omega )}{2}}{1+2\alpha {\left(\omega -\omega_{k}\right)}^{2}}$$

The center frequency update is as follows:(5)$$\omega_{k}^{n+1}=\frac{\sum_{p}\int_{0}^{\infty }\omega {\left|{\hat{u}}_{k,p}(\omega )\right|}^{2}d\omega }{\sum_{p}\int_{0}^{\infty }{\left|{\hat{u}}_{k,p}(\omega )\right|}^{2}d\omega }$$

Owing to the ability of the MVMD method to process multiple channel data simultaneously, the number of IMFs obtained from each channel’s decomposition is equal, and the center frequency is the same. This decomposition method makes signal analysis more stable and conducive to fault diagnosis.

#### 2.1.2. IMVMD

The modal number *K* and penalty factor *α* in the MVMD algorithm must be set manually, and these parameters directly affect the decomposition quality [30]. When *K* is extremely small, insufficient decomposition may occur, leading to mode mixing. Pseudo-components may appear when the K value is too large. The *α* value determines the bandwidth of the IMF component, which in turn affects the MVMD.

In this paper, a comprehensive performance indicator based on orthogonality and DE is proposed to obtain the appropriate MVMD parameters. The details of the indicator are as follows.

DE can detect time-series complexities and dynamic mutations. This method has the advantages of simple calculations, good robustness, and high computational efficiency.(6)$$DE(imf)=-\sum_{\pi =1}^{nc^{m}}p_{e}.\ln p_{e}$$
where *m* denotes the embedding dimension, *nc* is the number of classes, and *p_e_* is the probability of the *π*-th mode of the IMF. The smaller the DE value of the signal, the smoother the IMF curves.

The orthogonality index is mainly used to evaluate the independence between the two modes as follows:(7)$$O_{r}(a,b)=\frac{a\cdot b}{\left|a\right|\left|b\right|}$$
where **a** and **b** are two vectors. The lower the orthogonality value, the higher the independence of the two vectors. The larger the orthogonality index, the higher the possibility of modal mixing.

The better decomposition effect of MVMD is reflected in reduced modal mixing and smoother IMF curves, which means that the DE and orthogonality values are the minimum. Since the original signal is decomposed into several IMFs, corresponding to several DEs, the largest DE represents the worst IMF in the decomposition. If the worst IMF has a smaller DE, the remaining IMF curves will be smoother. Therefore, a comprehensive evaluation index is proposed in this study to assist in determining parameters *K* and *α*. The indicators are as follows:(8)$$\left\{\begin{array}{l} \min\;S_{y}(K,\alpha )=\hat{R}(K,\alpha )+\hat{D}(K,\alpha )\\ R(K,\alpha )=\frac{1}{C_{K}^{2}}\sum_{i,j=i+1}^{K}O_{r}(imf_{i},imf_{j})\\ D(K,\alpha )=\max\{DE(imf_{1}),DE(imf_{2}),\ldots ,DE(imf_{K})\} \end{array}\right.$$
where *imf* denotes the decomposition modal components of MVMD under control parameters *K* and *α*. $\hat{R}(K,\alpha )$ and $\hat{D}(K,\alpha )$ are the normalization forms of the values of R(K,α) and D(K,α) under different parameters. $C_{K}^{2}$ is the combination number.

The steps to obtain the parameters are as follows.

Step 1: The *S_y_* values for different values of *K* are calculated with a constant default value of *α*, and the *K* value corresponding to the minimum *S_y_* is obtained as the optimal value.

Step 2: The *S_y_* values for different values of *α* are calculated while keeping the optimal *K* value unchanged, and the *α* value corresponding to the minimum *S_y_* is obtained as the optimal value.

According to the above steps, the optimal parameters *K* and *α* can be obtained to improve the MVMD performance. The steps proposed in this study only require traversal calculations within the preset range of *K* and *α* according to the step size. The amount of calculation is usually small, and the results can generally be obtained within a few seconds of programming on an ordinary computer.

### 2.2. Ensemble Refined Composite Multivariate Multiscale Dispersion Entropy

#### 2.2.1. The Basic Theory of mvMDE

Multivariate multiscale dispersion entropy (mvMDE) was proposed by Azami in 2019 [21]. This method has a faster calculation speed, better stability, and fewer calculation parameters to be set and considered, making it suitable for the analysis of multivariate data.

Assume a multi-channel signal $U={\left\{u_{p,b}\right\}}_{b=1,2,\ldots ,L}^{p=1,2,\ldots ,P}$ with *P* channels and a data length of *L*. The calculation process for mvMDE is as follows.

(1) The coarse-graining process is used to obtain time series with different scales. The original time series is divided into non-overlapping segments of length *τ*. The mean value of each segment forms a new data series, as follows:(9)$$\left\{\begin{array}{l} x_{p,i}^{\tau }=\frac{1}{\tau }\sum_{b=(i-1)\tau +1}^{i\tau }u_{p,b}\;\\ where\;p=1,2,\ldots ,P\\ where\;i=1,2,\ldots ,(N=\left\lfloor \frac{L}{\tau }\right\rfloor ) \end{array}\right.$$
where *N* is the length of $x_{p}^{\tau }$, and *τ* is the scale factor.

(2) The multi-channel coarse-grained time series $X^{\tau }={\left\{x_{p,i}^{\tau }\right\}}_{i=1,2,\ldots ,N}^{p=1,2,\ldots ,P}$ is mapped to the range of 0 to 1 using a normal distribution function as follows:(10)$$y_{p,i}^{\tau }=\frac{1}{\sigma_{p}\sqrt{2\pi }}\int_{-\infty }^{x_{p,i}^{\tau }}e^{\frac{-{\left(t-\mu_{p}\right)}^{2}}{2\sigma_{p}^{2}}}dt$$
where *σ_p_*^2^ and *μ_p_* are the variance and mean value of time series *x^τ^_p_*, respectively.

(3) The series $Y^{\tau }={\left\{y_{p,i}^{\tau }\right\}}_{i=1,2,\ldots ,N}^{p=1,2,\ldots ,P}$ continues to be mapped to different categories as follows:(11)$$z_{p,i}^{\tau ,c}=R(c*y_{p,i}^{\tau }+0.5)$$
where $z_{p,i}^{\tau ,c}$ represents the *i*-th member of the classified signal in the *p*-th channel. *R* is the rounding function and *c* is the number of categories.

(4) The time series $z_{p,j}^{m,\tau ,c}$ is built with embedding dimension *m* and time delay *d*, as follows:(12)$$z_{p,j}^{m,\tau ,c}=\{z_{p,j}^{\tau ,c},z_{p,j+d}^{\tau ,c},z_{p,j+2d}^{\tau ,c}\ldots ,z_{p,j+(m-1)d}^{\tau ,c}\},\;j=1,2,\cdots ,N-(m+1)d$$

(5) Each $z_{p,j}^{m,\tau ,c}$ can be mapped to a dispersion pattern as(13)$$\pi_{v_{0}v_{1}\cdots v_{m-1}}\;,\;where\;z_{p,j}^{\tau ,c}=v_{0},z_{p,j+d}^{\tau ,c}=v_{1},\ldots ,z_{p,j+(m-1)d}^{\tau ,c}=v_{m-1}$$

Because the dispersion pattern contains *m* dimensions, each dimension can be one of the integers from 1 to *c*. Therefore, the number of dispersion patterns for $z_{p,j}^{m,\tau ,c}$ is *c^m^*.

(6) The relative frequency of each potential dispersion pattern of each channel is calculated according to Equation (13):(14)$$r(\pi_{v_{0}v_{1}\cdots v_{m-1}})=\frac{Number(\pi_{v_{0}v_{1}\cdots v_{m-1}}|z_{p,j}^{m,\tau ,c})}{(N-(m-1)d)P}$$
where $Number(\pi_{v_{0}v_{1}\cdots v_{m-1}}|z_{p,j}^{m,\tau ,c})$ represents the number of $z_{p,j}^{m,\tau ,c}$ mapped to the dispersion pattern $\pi_{v_{0}v_{1}\cdots v_{m-1}}$.

(7) The mvMDE value can be calculated according to Shannon entropy as(15)$$mvMDE(u,\tau ,m,c,d)=-\sum_{\pi =1}^{c^{m}}r(\pi_{v_{0}v_{1}\cdots v_{m-1}}).\ln r(\pi_{v_{0}v_{1}\cdots v_{m-1}}).$$

#### 2.2.2. ERCmvMDE

As stated above, equidistant segmentation and mean value methods are applied in the coarse-graining process for mvMDE. This processing method has the following limitations.

(I) The length of the time series after coarse-grained processing will decrease significantly with an increase in the scale factor. A shorter data length will decrease the stability and reliability of the mvMDE calculation. Meanwhile, this coarse-graining process has only one starting position, and the potential information in the time series processed by other starting positions is ignored.

(II) If the coarse-graining process uses only the mean value of the signal segments, some important information may be lost, such as the peaks and variances. This information is usually very important for the fault diagnosis of planetary gearboxes.

An ensemble refined composite multivariate multiscale dispersion entropy (ERCmvMDE) method is proposed in this paper to overcome the above limitations. Firstly, the sliding coarse-graining method is used instead of the equidistant segmentation method. The refined composite entropy calculation is more stable. Secondly, multiple forms of coarse-grained processing methods are adopted to extract more comprehensive fault feature information. The detailed steps of the ERCmvMDE method are as follows.

Step 1: Assuming a multi-channel signal with *P* channels and a data length of *L*,$U={\left\{u_{p,b}\right\}}_{b=1,2,\ldots ,L}^{p=1,2,\ldots ,P}$, and the scale factor is *τ*. The *τ* coarse-grained sequences are obtained as follows:(16)$$\left\{\begin{array}{l} X_{a}^{\tau }={\left\{x_{p,a,i}^{\tau }\right\}}_{i=1,2,\ldots ,N}^{p=1,2,\ldots ,P}\;\\ where\;a=1,2,\ldots ,\tau \;and\;i=1,2,\ldots ,(N=\left\lfloor \frac{L}{\tau }\right\rfloor ) \end{array}\right.$$

Four different coarse-grained methods are used to obtain the elements of $X_{a}^{\tau }$, as follows.

Coarse-grained processing by mean value:(17)$$x_{p,a,i}^{\tau }{|}_{mean}=\frac{1}{\tau }\sum_{b=(i-1)\tau +a}^{i\tau +a-1}u_{p,b}$$

Coarse-grained processing by root mean square amplitude (RMSA):(18)$$x_{p,a,i}^{\tau }{|}_{RMSA}={\left(\frac{1}{\tau }\sum_{b=(i-1)\tau +a}^{i\tau +a-1}\sqrt{\left|u_{p,b}\right|}\right)}^{2}$$

Coarse-grained processing by variance:(19)$$x_{p,a,i}^{\tau }{|}_{variance}=\frac{1}{\tau }\sum_{b=(i-1)\tau +a}^{i\tau +a-1}{\left(u_{p,b}-\frac{1}{\tau }\sum_{b=(i-1)\tau +a}^{i\tau +a-1}u_{p,b}\right)}^{2}$$

Coarse-grained processing by maximum value:(20)$$x_{p,a,i}^{\tau }{|}_{max}=\underset{(i-1)\tau +a\leq b\leq i\tau +a-1}{\max}\{u_{p,b}\}$$

Using the above four coarsening strategies, four types of coarsening sequence groups can be independently calculated as $X_{a}^{\tau }{|}_{mean}$, $X_{a}^{\tau }{|}_{RMSA}$, $X_{a}^{\tau }{|}_{variance}$, and $X_{a}^{\tau }{|}_{max}$.

Step 2: For groups of coarse-grained sequences of the same type, the relative frequency of each potential dispersion pattern for each coarse-grained sequence is calculated according to Equations (10)–(14), and the average relative frequency of each potential dispersion pattern for all different starting points is calculated as follows:(21)$$\bar{r}(\pi_{v_{0}v_{1}\cdots v_{m-1}})=\frac{1}{\tau }\sum_{a=1}^{\tau }r(\pi_{v_{0}v_{1}\cdots v_{m-1}}\left|X_{a}^{\tau }\right.)$$
where $r(\pi_{v_{0}v_{1}\cdots v_{m-1}}\left|X_{a}^{\tau }\right.)$ represents the relative frequency of the dispersion patterns of the *a*-th coarse-grained sequence.

Step 3: The RCmvMDE value can be calculated according to Shannon entropy as(22)$$RCmvMDE(u,\tau ,m,c,d)=-\sum_{\pi =1}^{c^{m}}\bar{r}(\pi_{v_{0}v_{1}\cdots v_{m-1}}).\ln\bar{r}(\pi_{v_{0}v_{1}\cdots v_{m-1}}).$$

Step 4: The RCmvMDE values calculated with the different coarse-graining strategies are combined to obtain the ERCmvMDE, as follows:(23)$$\begin{array}{l} ERCmvMDE(u,\tau ,m,c,d)=\{RCmvMDE_{mean}(u,\tau ,m,c,d),\ldots \\ \;\ldots ,RCmvMDE_{max}(u,\tau ,m,c,d),\ldots \\ \;\ldots ,RCmvMDE_{RMSA}(u,\tau ,m,c,d),\ldots \\ \;\ldots ,RCmvMDE_{variance}(u,\tau ,m,c,d)\} \end{array}$$

The ERCmvMDE contains the RCmvMDE when using mean value coarsening strategies.

## 3. The Structure of the Proposed Method

In this study, a fault diagnosis method based on IMVMD, ERCmvMDE, and the LSSVM is proposed for planetary gearboxes. The fault diagnosis flowchart is shown in Figure 1.

The detailed steps of the proposed methods are as follows.

Step 1: The appropriate parameter values of *K* and *α* are obtained by the proposed IMVMD, and the multi-channel signal of the planetary gearbox is decomposed to obtain the IMFs.

Step 2: The ERCmvMDE values of the IMFs are calculated and used as feature vectors for fault diagnosis.

Step 3: The LSSVM is trained using feature vectors, and the trained LSSVM is used to classify the predicted samples for fault diagnosis.

## 4. Simulation Analyses

In this section, the performance of the IMVMD and ERCmvMDE methods proposed in this study is verified through simulations.

### 4.1. Simulation Analysis of IMVMD

Multi-channel observation signals with multiple characteristic frequencies are considered as *X*(*t*) = {*X*_1_(*t*) + *n_G_*(*t*), *X*_2_(*t*) + *n_G_*(*t*)}, where *n_G_*(*t*) is Gaussian white noise. The signal to noise ratio (SNR) is set to −5 dB, and the observation signals can be described as follows:(24)$$\left\{\begin{array}{l} x_{1}(t)=\cos(2\pi *10t)+0.5*\cos(2\pi *20t)+0.25*\cos(2\pi *30t)\\ x_{2}(t)=2*\cos(2\pi *10t+\pi /2)+1.5*\cos(2\pi *20t+\pi /2)+\cos(2\pi *30t+\pi /2) \end{array}\right.$$

The multi-channel signal contains components of 10 Hz, 20 Hz, and 30 Hz, which are similar to the basic frequency and multiple frequencies of a planetary gearbox. The observed waveforms are shown in Figure 2.

The optimal parameters *K* and *α* are obtained using the proposed IMVMD. The *S_y_* values for different values of *K*∈[2, 10] are calculated using a constant default value of *α* = 2000. The results are shown in Figure 3.

It can be seen that the *S_y_* value reaches a minimum when *K* is 3, which means that the IMFs are the most regular and smooth and are not easily mixed up. The decomposition performance of MVMD is best when *K* is equal to 3. There are just three frequency components in the original signals according to Equation (24), which indicates that IMVMD has obtained the optimal *K*.

The *S_y_* values for different values of *α*∈[100, 5000] are calculated using a constant default value of *K* = 3, where the step of *α* is 100. The results are shown in Figure 4.

As shown in Figure 4, it can be seen that when *α* is 3600, the fitness function has a minimum value. Therefore, the optimal parameters obtained by IMVMD are *K* = 3 and *α* = 3600. The final decomposition effects of IMVMD are shown in Figure 5.

As shown in Figure 5, both channel signals are decomposed into IMFs that are smooth and regular. As shown in the frequency spectra of the IMFs, the center frequencies of the IMFs of the two-channel signals are strictly related to the three characteristic frequencies of the original signal. As shown in Figure 5a, the 30 Hz component of *x*_1_ almost disappears under strong noise interference, but the IMF with a 30 Hz center frequency is still decomposed by IMVMD. The results indicate that IMVMD has excellent performance in signal decomposition.

### 4.2. Simulation Analysis of ERCmvMDE

To verify the performance of the proposed ERCmvMDE, multi-channel signals formed by power-law noise and white noise are applied. Power-law noise is colored noise based on white noise, and it is difficult to distinguish between white noise and power-law noise directly in the time domain.

Signals with three channels are applied in this section. As the signals are formed by power-law noise and white noise, four types of multi-channel signals are formed with three independent channel signals as follows: (1) all three channels contain white noise; (2) two channels contain white noise and one channel contains power-law noise; (3) one channel contains white noise and two channels contain power-law noise; (4) all three channels contain power-law noise. The length of the multi-channel signal is 1024, and each type of data is independently generated into 20 groups.

The RCmvMSE [21], RCmvMFE [26], RCmvMDE [27], and ERCmvMDE methods are used to calculate the entropy of the multi-channel signals. The parameters for the different methods are listed in Table 1. The mean values and standard deviations (SDs) of the results obtained by the different methods are shown in Figure 6.

Figure 6a–c show the mean values and SDs of RCmvMFE, RCmvMSE, and RCmvMDE, respectively. Figure 6c–f show the mean values and SDs of the results for ERCmvMDE. RCmvMFE and RCmvMSE have larger SDs, particularly for signals with more channels of power-law noise. RCmvMDE and ERCmvMDE have small SDs, which means that these methods have good computational stability. As shown in Figure 6a,b, the entropies of the four types of signals cannot be distinguished from each other well. However, the ERCmvMDE values of the four types of signals can be clearly distinguished from each other.

The ERCmvMDE values based on the mean value, maximum value, and RMSA decrease with an increase in the scale factor, and there is a more significant decrease if the signal contains more white noise channels. However, the ERCmvMDE values based on variance show the opposite trend. This indicates that ERCmvMDE can obtain more useful information than RCmvMDE.

The results indicate that ERCmvMDE not only has good computational stability but can also extract rich and effective feature information.

## 5. Experimental Analyses

### 5.1. Description of the Case

A public failure dataset for a planetary gearbox, named the ‘WT-Planetary Gearbox Dataset’, provided by Liu et al. [31], was used to verify the proposed method. The failure data were produced by a drive train dynamic simulator to simulate the vibration data of a planetary gearbox of a wind turbine. The test rig included a planetary gearbox, motor, fixed-shaft gearbox, loads, and data collection device. The planetary gearbox consisted of four planet gears and a sun gear, and the detailed parameters of the planetary gearbox are listed in Table 2. The planetary gearbox was operated under five different conditions: healthy, broken tooth of sun gear, wear gear of sun gear, root crack of sun gear, and missing tooth of sun gear. The vibration data in the two directions, i.e., X and Y, were collected at a sampling frequency of 48 k Hz.

The time-domain waveforms of the vibration data under different operating conditions are shown in Figure 7.

Because the dataset had sufficiently long data samples and the collection time for each condition exceeded 5 min, the dataset contained over 14 million data points for each operation condition. The dataset for each operation condition was evenly segmented into 200 groups, and each group contained 72,000 data points. Then, a sample of 1024 consecutive points with random starting positions was selected for each group, and a non-overlapping sample dataset was built for analysis in this study. Because of the use of the LSSVM for sample classification in this study, the dataset had to be divided into a training set and a testing set. The detailed parameters of the dataset are listed in Table 3.

### 5.2. Feature Extraction Performance

The vibration dataset of the planetary gearbox was processed by IMVMD, and the optimal parameters obtained by IMVMD were *K* = 4 and *α* = 140. The vibration datasets were decomposed by IMVMD with the optimal parameters. Then, the IMFs were processed by ERCmvMDE to extract features, and RCmvMSE, RCmvMFE, and RCmvMDE were also applied to extract features for comparison. The mean values and SDs of the results are shown in Figure 8.

Figure 8a–c show the mean values and SDs of RCmvMFE, RCmvMSE, and RCmvMDE, respectively. Figure 8c–f show the mean values and SDs of the results for ERCmvMDE. The mean values of RCmvMSE, RCmvMFE, and RCmvMDE for IMF2 to IMF4 were less smooth as the scale factor increased, and the mean values of the different operating conditions were significantly mixed. It was difficult to achieve effective fault diagnosis. As shown in Figure 8d–f, the mean values of each IMF had better continuity as the scale factor increased, and the distance between each operating condition increased significantly, which suppressed the mixing problem. These results indicate that the features extracted by ERCmvMDE are more conducive to fault diagnosis.

As the feature extraction methods are based on the multiscale entropy of the IMFs, the dimensions of the features are always too large, which makes intuitive judgment difficult. Thus, the t-Distributed Stochastic Neighbor Embedding (t-SEN) method was applied to enable the multi-dimensional features to be displayed in a 2D space. The t-SEN visualizations of the different feature extraction methods are shown in Figure 9.

As shown in Figure 9, the sample features based on RCmvMSE and RCmvMFE between different operating conditions were too close, which is not conducive to classification. The sample feature of RCmvMDE appeared to exhibit a clustering phenomenon, but some of them appeared to be mixed. The sample feature based on ERCmvMDE appeared clustered for different operation conditions, but the features of each operation condition could be separated well. In order to quantitatively illustrate the effects of the different feature extraction methods, the within-class scatter, between-class scatter, and a separability index were used to measure the clustering and classification effects of the feature datasets [32]. The between-class scatter describes the classification effect between samples of different classes, while the within-class scatter describes the clustering effect between samples of the same class. The calculation of these indices is as follows:(25)$$\begin{array}{l} \left\{\begin{array}{l} S_{b}=\frac{2}{C(C-1)}\sum_{k=1}^{C-1}\sum_{l=k+1}^{C}\left\|{\bar{y}}_{k}-{\bar{y}}_{l}\right\|\\ S_{w}=\frac{1}{C}\frac{1}{n}\sum_{k=1}^{C}\sum_{i=1}^{n}\left\|y_{ki}-{\bar{y}}_{k}\right\|\\ \gamma =\frac{S_{b}}{S_{w}}\\ {\bar{y}}_{k}=\frac{1}{n}\sum_{i=1}^{n}y_{ki} \end{array}\right.\\ y_{ki}(k=1,2,\ldots ,C;i=1,2,\ldots ,n) \end{array}$$
where $S_{b}$ is the between-class scatter, $S_{w}$ is the within-class scatter, γ represents the separability index, $y_{ki}(k=1,2,\ldots ,C;i=1,2,\ldots ,n)$ is the dataset of features, C is the number of classes, and n is the sample number. The larger $S_{b}$ and γ are and the smaller $S_{w}$ is, the better the clustering and classification effects of the feature set.

The evaluation indices of the features extracted by the different methods are listed in Table 4.

The results indicate that the proposed method based on IMVMD and ERCmvMDE has better performance in the feature extraction of the wind turbine planetary gearbox.

### 5.3. Fault Diagnosis Analyses

The LSSVM was applied for fault classification; the training sample set, test sample set, and sample labels were set as shown in Table 3.

#### 5.3.1. Performance of IMVMD in Fault Diagnosis

To verify the performance of IMVMD in fault diagnosis, the IMFs were obtained with the optimal parameters (*K* = 4, *α* = 140) and default parameters (*K* = 5, *α* = 200). The ERCmvMDE of the IMFs was then calculated and formed as a feature vector for the LSSVM. Confusion diagrams of the fault diagnosis results are shown in Figure 10.

The results indicated that the fault diagnosis accuracy based on the optimal parameters obtained by IMVMD was significantly improved. The fault diagnosis accuracy can reach 99.5% when using the method proposed in this study. The results further demonstrate that IMVMD can improve the decomposition efficiency.

#### 5.3.2. Performance of ERCmvMDE in Fault Diagnosis

To verify the performance of ERCmvMDE in fault diagnosis, feature extraction methods based on different entropies were compared for fault diagnosis. The IMFs were obtained by IMVMD, and the RCmvMSE, RCmvMFE, RCmvMDE, and ERCmvMDE of the IMFs were calculated and formed as a feature vector for the LSSVM. Confusion diagrams of the fault diagnosis results are shown in Figure 11.

As shown in Figure 11, the fault diagnosis accuracies based on RCmvMSE, RCmvMFE, RCmvMDE, and ERCmvMDE were 96.5%, 97.75%, 97.5%, and 99.5%, respectively. Feature extraction based on ERCmvMDE provided better fault diagnosis than the other feature extraction methods. The fault diagnosis based on RCmvMSE resulted in incorrect results for the broken tooth, crack, and wear gear conditions. The fault diagnosis based on RCmvMFE resulted in some incorrect results in almost all conditions, except for the missing tooth condition. Fault diagnosis based on RCmvMDE resulted in incorrect results under almost all conditions except the wear gear condition. The proposed fault diagnosis method resulted in fewer incorrect results, and there were only two mistakes in the broken tooth and crack conditions.

To validate the adaptability of the proposed method across different data samples, a random starting point approach was employed to acquire diverse samples. As stated above, the original data were evenly segmented into 200 groups with 72,000 data points for each state, and the sample data length was only 1024. Therefore, a random starting point could form diverse data samples. The data sample sets were randomly generated 20 times, the IMFs were obtained by IMVMD, and the RCmvMSE, RCmvMFE, RCmvMDE, and ERCmvMDE of the IMFs were calculated and formed as a feature vector for the LSSVM. The average fault diagnosis accuracies are listed in Table 5.

The results indicate the proposed methods still had good performance with different data sample sets.

The above experimental analysis results indicate that the proposed IMVMD and ERCmvMDE can make feature extraction more effective, and the proposed fault diagnosis method performs well in the fault diagnosis of wind turbine planetary gearboxes.

## 6. Conclusions

Feature extraction from multi-channel vibration signals is an important issue for the fault diagnosis of wind turbine planetary gearboxes. A fault diagnosis method based on a novel feature extraction method was proposed in this study. Firstly, an improved MVMD was proposed to obtain the optimal parameters of the MVMD, which would make the MVMD more effective. Secondly, ERCmvMDE was proposed to extract rich and effective feature information. Finally, the LSSVM was applied for fault classification. The following conclusions were drawn from the simulations and experimental studies.

(a) IMVMD can obtain the optimal parameters of *K* and *α* of MVMD, which will affect the decomposition effect. The fault diagnosis accuracy when using IMVMD was higher than that using MVMD.

(b) ERCmvMDE not only exhibits good computational stability but can also extract rich and effective feature information.

(c) The fault diagnosis accuracy using ERCmvMDE was higher than that using RCmvMSE, RCmvMFE, and RCmvMDE in the experimental study. In additional, testing work needs to be conducted to verify the broad applicability of this method in wind turbine planetary gearbox fault diagnosis in the future.

## Figures and Tables

**Figure 1 entropy-27-00192-f001:**
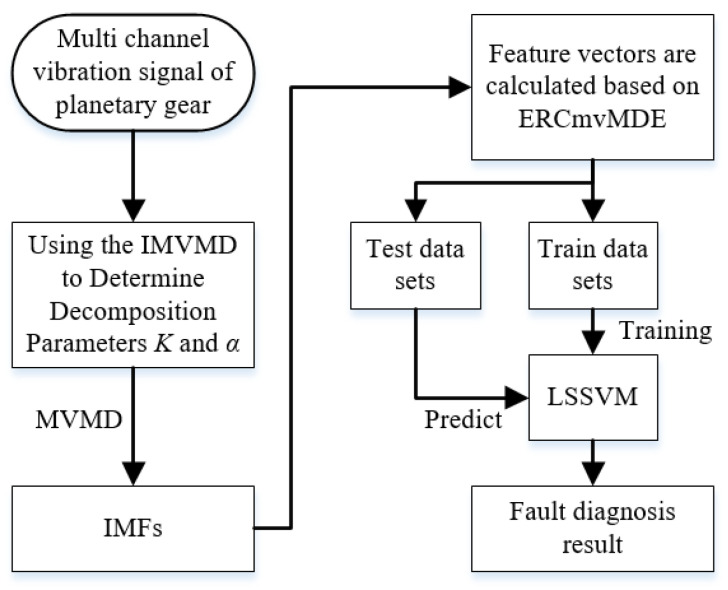
The flowchart of the proposed fault diagnosis method.

**Figure 2 entropy-27-00192-f002:**
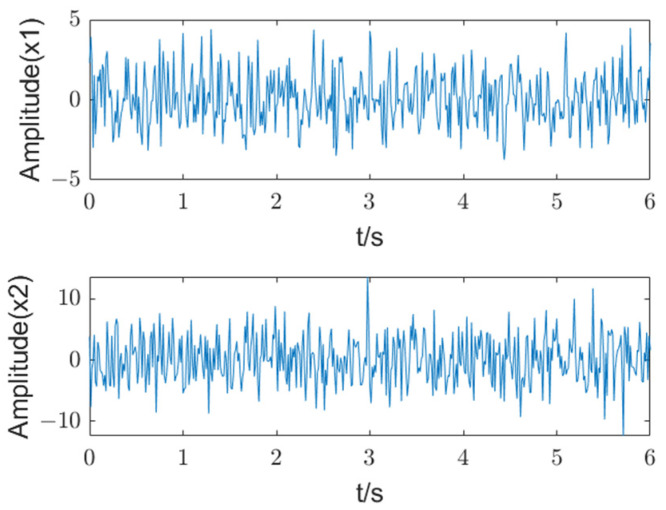
The time-domain waveforms of a multi-channel signal.

**Figure 3 entropy-27-00192-f003:**
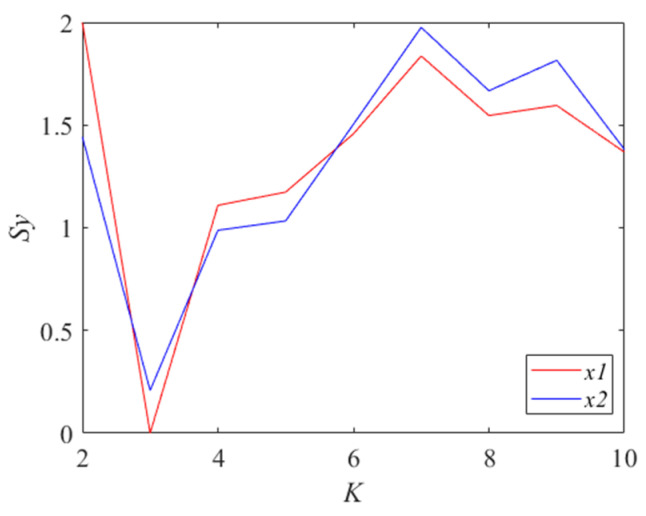
The Sy values for different *K* (*α* = 2000).

**Figure 4 entropy-27-00192-f004:**
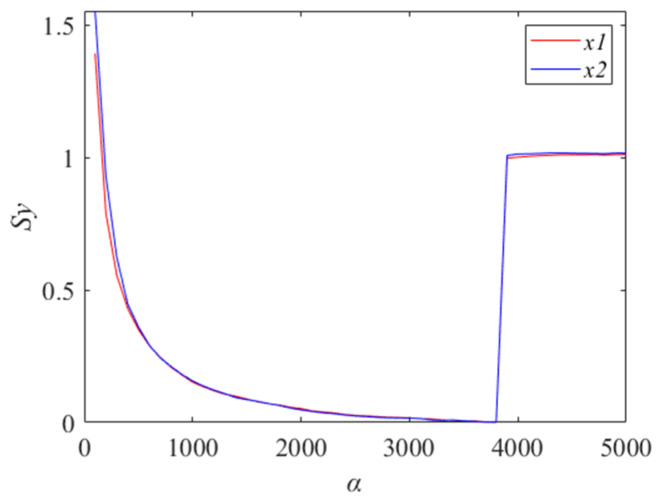
The Sy values for different *α* (*K* = 3).

**Figure 5 entropy-27-00192-f005:**
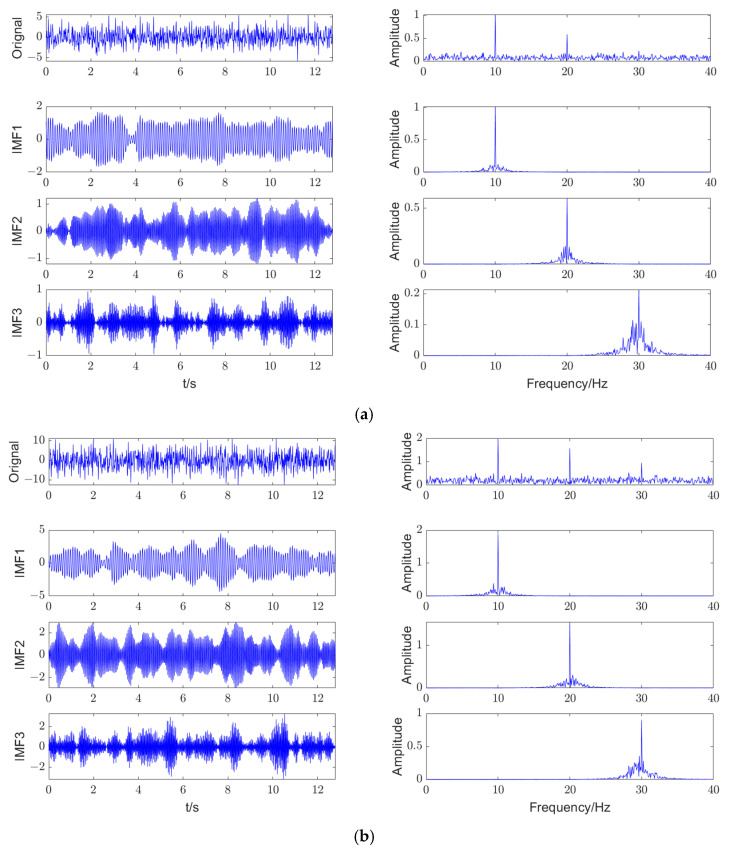
The final decomposition effects of IMVMD. (**a**) *x*_1_, (**b**) *x*_2_.

**Figure 6 entropy-27-00192-f006:**
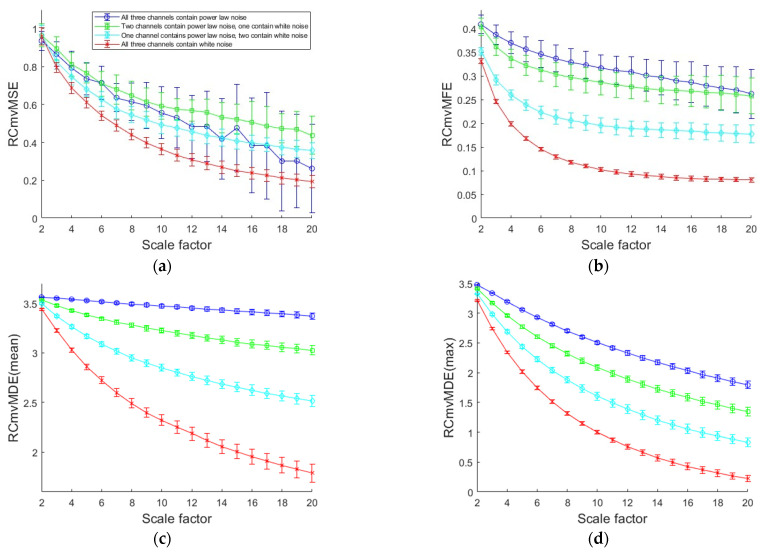
The mean values and SDs of the results obtained by different methods with 20 groups of data. (**a**) RCmvMSE; (**b**) RCmvMFE; (**c**) RCmvMDE (mean); (**d**) RCmvMDE (maximum); (**e**) RCmvMDE (variance); (**f**) RCmvMDE (RMSA).

**Figure 7 entropy-27-00192-f007:**
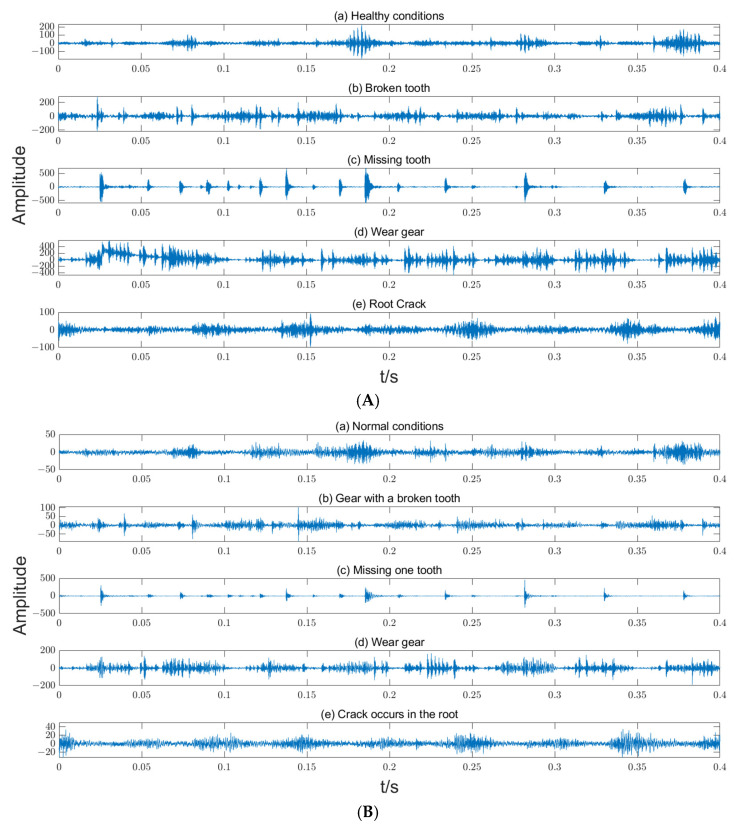
The time-domain waveforms of the vibration data of the planetary gearbox. (**A**) X direction; (**B**) Y direction.

**Figure 8 entropy-27-00192-f008:**
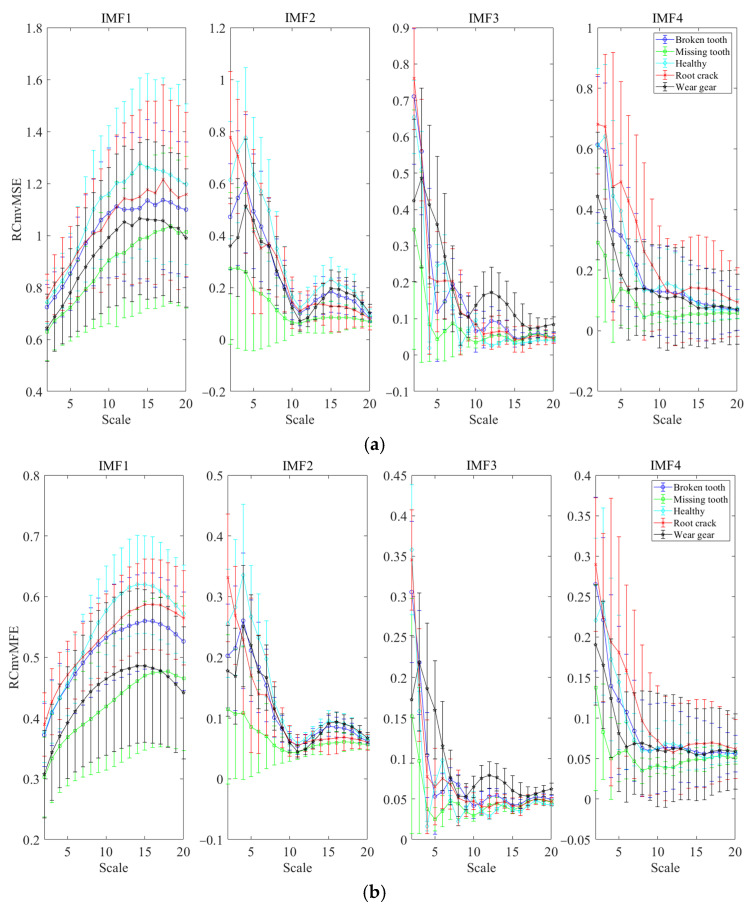
The mean values and SDs of the results obtained by different methods with 200 groups of sampling data. (**a**) RCmvMSE; (**b**) RCmvMFE; (**c**) RCmvMDE (mean); (**d**) RCmvMDE (maximum); (**e**) RCmvMDE (variance); (**f**) RCmvMDE (RMSA).

**Figure 9 entropy-27-00192-f009:**
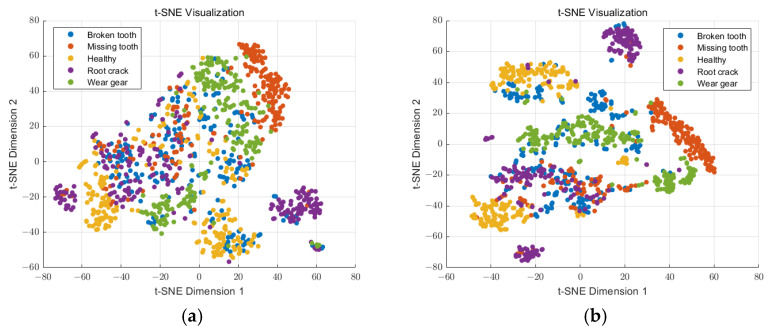
The t-SEN visualizations of different feature extraction methods. (**a**) IMVMD+ RCmvMSE, (**b**) IMVMD + RCmvMFE, (**c**) IMVMD + RCmvMDE, (**d**) IMVMD + ERCmvMDE.

**Figure 10 entropy-27-00192-f010:**
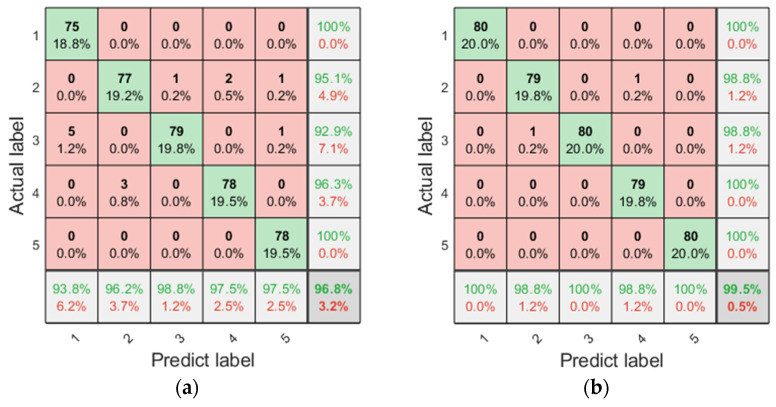
The confusion diagram of fault diagnosis with different parameters. (**a**) With default parameters (*K* = 5, *α* = 2000); (**b**) with optimal parameters (*K* = 4, *α* = 140).

**Figure 11 entropy-27-00192-f011:**
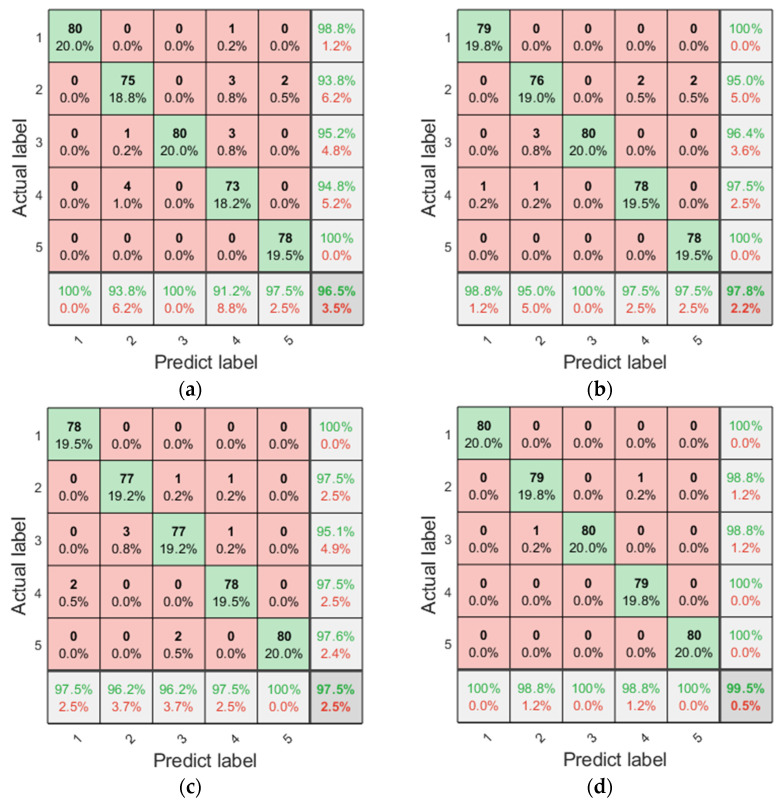
The confusion diagram of fault diagnosis with different feature extraction methods. (**a**) RCmvMSE; (**b**) RCmvMFE; (**c**) RCmvMDE; (**d**) ERCmvMDE.

**Table 1 entropy-27-00192-t001:** The parameter settings for the different methods.

Method	Unique Parameters	Common Parameters
ERCmvMDE	Number of classes is 6	Scale factor is 20 Scalar embedding value is 2 Scalar time lag value is 1
RCmvMDE	Number of classes is 6	
RCmvMSE	Scalar threshold value is 0.15	
RCmvMFE	Scalar threshold value is 0.15 Fuzzy power is 2	

**Table 2 entropy-27-00192-t002:** The parameters of the planetary gearbox.

Rotating Frequency of Sun Gear	Tooth Number		
*f_r_* = 20 Hz	Sun gear	Ring gear	Planet gear
	28	100	36

**Table 3 entropy-27-00192-t003:** Parameters of sample dataset.

Operation Condition	Data Length	Number of Train Samples	Number of Test Samples	Label
Healthy	1024	120	80	1
Broken tooth				2
Missing tooth				3
Root crack				4
Wear gear				5

**Table 4 entropy-27-00192-t004:** The evaluation indices of the features extracted by the different methods.

Method	$S_{b}$	$S_{w}$	γ
IMVMD+ RCmvMSE	29.109	37.130	0.784
IMVMD + RCmvMFE	25.297	38.374	0.660
IMVMD + RCmvMDE	40.459	38.117	1.060
IMVMD + ERCmvMDE	55.202	27.857	1.982

**Table 5 entropy-27-00192-t005:** Average fault diagnosis accuracies of different methods when applied 20 times.

Method	RCmvMSE	RCmvMFE	RCmvMDE	ERCmvMDE
Average accuracy	96.44%	97.18%	96.94%	99.31%

## Data Availability

The data that support the findings of this study are available upon request from the authors.

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
