# Peer review of "Fault Diagnosis of Wind Turbine Gearbox Based on Improved Multivariate Variational Mode Decomposition and Ensemble Refined Composite Multivariate Multiscale Dispersion Entropy"

_entropy, 2025, doi:10.3390/e27020192_

Round 1
Reviewer 1 Report
Comments and Suggestions for Authors
Please referring to the attachment.

Reviewer 2 Report
Comments and Suggestions for Authors
Thanks for this interesting research. The topic that you are dealing with is of high interest in the mechanical community, where the diagnosis of planetary gearboxes is gaining relevance significantly in the last years.
The main contributions of this manuscript lie in a new method for gearbox failure diagnosis that involves using a comprehensive evaluation index based on orthogonality and dispersion entropy to obtain the optimal parameters of MVMD, and combines such with an ERCmvMDE to form the composite feature.
Main suggestions (further details can be extracted from the commented manuscript, where RED and YELLOW marking is using for comments, the GREEN marking is used only to facilitate my understanding and is not connected to any comments or recommendations):
- Be attentive to defining abbreviations before using them
- The introduction fails to provide a clear context for the constrained applicability of the simulation and empirical validations. In the simulation and test sections, this becomes even more evident: even if the method's applicability is clearly broad, your actual validation (simulation and test) is constrained to certain parameters without providing a clear justification for these.
- The test part requires more background information on the test specimen. Further, a description of the actual IMFs in equation form would potentially allow to link these to the actual configuration of the gearbox, which would be a valuable insight worth exploring, in my opinion.
- For the rest, just some typos and including some restrictions to the last statement in your conclusions.
In general, the manuscript and the research is highly interesting and just requires some fine-polishing in terms of the scientific soundness and about making it easier to understand for a gearbox engineer that is not specialised in these mathematical methods, in my opinion.
